# Development, implementation and evaluation of the online Movement, Interaction and Nutrition for Greater Lifestyles in the Elderly (MINGLE) program: The protocol for a pilot trial

**Diana Tang**[1]*, **Rona Macniven**[1,2], **Nicholas Bender**[3], **Charlotte Jones**[4], **Bamini Gopinath**[1]

1 Macquarie University Hearing, Macquarie University, North Ryde, New South Wales, Australia, 2 School of Population Health, Faculty of Medicine and Health, UNSW Sydney, Kensington, New South Wales, Australia, 3 Middlesex University, The Burroughs, London, United Kingdom, 4 Southern Medical Program, University of British Columbia, Okanagan Campus, Kelowna, Canada

* d.tang@mq.edu.au

**Data Availability Statement:** No datasets were generated or analysed during the current study. All

## Abstract

### Introduction

People with age-related macular degeneration (AMD) are more likely to experience loneliness, have poorer diets and be less physically active than people without AMD. The online Movement, Interaction and Nutrition for Greater Lifestyles in the Elderly (MINGLE) program is a holistic evidence-based intervention aiming to support people with AMD by incorporating physical activity, social interaction and nutrition education components all delivered via a COVID-19-safe Zoom platform. This study will involve two phases: 1) a formative qualitative study with AMD patients to identify the barriers and facilitators to participating in the proposed MINGLE program; and 2) a 10-week pilot study to evaluate the feasibility, acceptability and preliminary efficacy of MINGLE.

### Methods and analysis

Phase 1 involves AMD patients who will be recruited from an eye clinic in Western Sydney, Australia to participate in audio-recorded semi-structured interviews. Verbatim interview transcripts will be coded using the Capability, Opportunity, Motivation and Behaviour (COM-B) model and themes established. These themes will be used as a guide to specifically tailor the proposed MINGLE program to people with AMD. Phase 2 involves 52 AMD patients who will then be recruited from the same clinic to participate in the MINGLE program. Pre-post questionnaires will be administered to intervention participants to collect information on the following variables: demographics, socioeconomic status, vision function, loneliness, quality of life (including depression), falls risk, physical activity (level), and dietary intake. The acceptability and feasibility of the MINGLE program will also be evaluated using descriptive statistics.

relevant data from this study will be made available upon study completion.

**Funding:** D.T, R.M, C.J, and B.G were awarded the Macular Disease Foundation Australia Research Grant by the Macular Disease Foundation Australia to conduct this research project. https://www.mdfoundation.com.au/ The funders had and will have a role in the study design, data collection and analysis, decision to publish or preparation of the manuscript.

**Competing interests:** The authors have declared that no competing interests exist.

## Trial registration number

ACTRN12621000939897p.

## Introduction

Age-related macular degeneration (AMD) is the leading cause of vision loss and blindness in high income countries with an expected global prevalence of approximately 300 million by 2040 [1]. Vision loss also has wider health impacts and has been associated with higher rates of loneliness, reduced quality-of-life, and increased falls risk [2–4]. Among older adults in particular, the research literature shows that loneliness is more prevalent in visually impaired elderly than normal sighted elderly [3, 4]. Examples of this include a Dutch study which reported a significant difference of 21% more visually impaired older adults experiencing feelings of loneliness compared to normal sighted older adults (50% vs. 29%, respectively) [3]. Another study in Norway showed that among adults aged $\geq$ 66 years, there were 8.7% and 8.6% more visually impaired elderly suffering from moderate and severe loneliness, respectively, compared to the general Norwegian population of elderly [4].

Feelings of loneliness have also been associated with depressive symptoms [5], where the latter is already prevalent in up to 63% of visually impaired older adults including individuals experiencing minimal vision loss [2]. Moreover, persistent loneliness can also lead to poorer lifestyle behaviours [6, 7], which is particularly concerning for people with AMD, as growing research evidence supports an association between lifestyle risk factors such as a poor diet and low levels of physical activity with an increased risk of AMD development and/or progression [8–10]. Persistent loneliness has also been linked to other health outcomes including dementia and cardiovascular health risks [6, 7, 11, 12]. Therefore, there is a strong need for evidence-based intervention programs focused on reducing loneliness in visually impaired older adults, such as those diagnosed with AMD.

Lifestyle interventions that include education, behaviour change and/or group support components can be effective in reducing loneliness and depression in older adults [13, 14]. One example of an evidence-based intervention is the 'Walk N' Talk for your Life (WTL) program that was developed by our group in 2014 for low-income older adults in Canada [15]. This 12-week WTL program is ongoing and includes group walking; resistance training and balance exercises; and discussion of health topics [15]. Since its inception in 2014, more than 300 older adults have participated in this community-based, student and volunteer-run program across multiple locations [15]. With the success of WTL, we also developed a virtual adaptation of WTL hosted over Zoom (WTL-Z) [Zoom Communications Inc. 2016] to address the safety concerns related to COVID-19, particularly amongst vulnerable populations such as older adults. In 2021, a mixed methods randomised controlled trial (RCT) of WTL-Z involving 75 older adults in the United Kingdom was conducted as part of an unpublished Master of Research dissertation [Bender et al., Walk and Talk for Your Life: hosted over Zoom: A mixed methods study on the effects of an online, videoconference-based group exercise and health discussion intervention on mental health in older adults affected by COVID-19 social distancing restrictions]. After 10-weeks, participants who underwent the WTL-Z intervention (n = 35) reduced their feelings of loneliness by 25% and depressive symptoms fell by 30% compared to the control group (n = 40) who's baseline measurements did not improve significantly over this time. Initial qualitative analysis indicated WTL-Z was feasible and well-received by the participants. Other adaptations of WTL include the 'Walk, Talk and Listen'

pilot randomised controlled trial targeting functional fitness and loneliness in older adults with hearing loss (n = 66) [16]. This intervention adapted the successful socialisation and exercise components of WTL and incorporated group auditory rehabilitation to educate participants about hearing and available supporting devices, improve communication skills and provide psychosocial support [16]. Walk, Talk and Listen showed significant improvement in emotional and social loneliness and hearing-related quality of life in relation to group auditory rehabilitation attendance and poorer baseline hearing-related quality of life [16].

Amongst people with AMD, positive participant responses to virtual programs have also been reported in another evidence-based intervention developed by our group, which aimed to improve the dietary intakes of people with AMD through telehealth nutrition intervention [17]. This telehealth intervention involved the provision of an evidence-based workbook on nutrition and AMD links, together with monthly individual phone calls with a dietitian over four months, which significantly improved the participants' dietary intake of nutrient-rich dark green leafy vegetables and legumes and reduced consumption of packaged and processed foods [17].

To address the impacts of vision loss caused by AMD, including the increased prevalence of loneliness, depression and falls risk, as well as to improve AMD-related lifestyle risk factors (poor diet and physical inactivity), we aimed to build on our prior research by developing a holistic program called MINGLE (Movement, Interaction and Nutrition for Greater Lifestyles in the Elderly) and piloting this novel program in a sample of older Australian adults with AMD. Therefore, this study aims to: 1) develop the MINGLE program tailored to AMD patients, by incorporating the successful aspects of existing evidence-based interventions developed by our group (WTL-Z and a dietary intervention [17]); and 2) deliver the MINGLE program online via Zoom [18] in a non-randomised group of AMD patients. Although our prior research has shown that online delivery of WTL-Z can effectively reduce loneliness and depression, this study was conducted in normal sighted older adults residing in the UK. Hence, this pilot study is needed to evaluate the acceptability and feasibility of delivering the online MINGLE program in visually impaired older adults. The aims of this study will be achieved through two phases. Phase 1 will conduct semi-structured interviews with AMD patients to identify the barriers and facilitators to participating in the proposed MINGLE program; to determine the participants' thoughts about MINGLE and what they would change or would like included in the program and to accordingly modify the proposed MINGLE program to better suit the needs of people with AMD. Phase 2 will then involve a non-randomised pilot trial of the modified MINGLE program in AMD patients to determine the program's acceptability and feasibility and its preliminary efficacy.

## Materials and methods

### Recruitment

This study includes a qualitative component (semi-structured interviews) and a pilot trial of the MINGLE program (Fig 1). From October 2021, participants for both phases will be recruited by a research team member from a private eye clinic in Western Sydney, Australia. Recruitment processes will follow state-based coronavirus-19 (COVID-19) safety recommendations and thus, in the case of restrictions to in-person recruitment, participants will be recruited via telephone. Clinic staff will assist the researcher with telephone-based recruitment for both phases by providing the researcher with the contact details of eligible participants who have expressed an interest in participating in research projects. A flyer advertising each phase of the study will also be placed at the clinic for patients who would like to contact the researcher directly about participation in the study.

| | STUDY PERIOD | | | |
|---|---|---|---|---|
| | **Enrolment** | **Allocation** | **Post-allocation** | **Close-out** |
| **Timepoint** | *-t* | *0* | $t_1 - t_{10}$ | $t_{>10}$ |
| **Phase 1** | | | | |
| **Enrolment:** | | | | |
| *Eligibility screen* | X | | | |
| *Informed consent* | X | | | |
| **Semi-structured interviews** | | X | | |
| **Phase 2** | | | | |
| **Enrolment:** | | | | |
| *Eligibility screen* | X | | | |
| *Informed consent* | X | | | |
| *Allocation* | | X | | |
| **Intervention:** | | | | |
| *10-week MINGLE program* | | | X | |
| **Assessments:** | | | | |
| *Active Australia Survey* | X | | | X |
| *Short Dietary Questionnaire* | X | | | X |
| *National Eye Institute Visual Function Questionnaire* | X | | | X |
| *Falls Self-Efficacy Scale* | X | | | X |
| *De Jong Gierveld Loneliness Scale* | X | | | X |
| *Assessment of Quality of Life-8 Dimensions* | X | | | X |
| *Attitude to Falls-Related Intervention Scale* | X | | | X |
| *Feedback Questionnaire* | | | | X |

**Fig 1. MINGLE pilot trial schedule of enrolment, interventions, and assessments.**

**Phase 1.** Participants for the semi-structured interviews will be recruited over three months or until data saturation is achieved. Interviews will be conducted in-person in a private room at the eye clinic or over the telephone depending on the participant's preference and/or COVID-19 restrictions at the time. The inclusion criteria for Phase 1 are: 1) diagnosis of any form of AMD, 2) fluent English and 3) consent to participate in the study (including audio recording). Patient eligibility and interests will be screened by the researcher who will provide an explanation of the study. Informed verbal consent will be audio recorded and documented in a Record of Verbal Consent form.

**Phase 2.** Recruitment for Phase 2 will commence after the preliminary findings from Phase 1 have been identified. These findings will help to inform any modifications to the MINGLE program prior to conducting the pilot study. Participants will be recruited over six months or until the sample size target of 52 AMD patients is achieved. The inclusion criteria for Phase 2 are: 1) diagnosis of any form of AMD, 2) fluent English, 3) access to a smart phone, tablet or laptop with a front-facing camera; 4) technical ability to use Zoom or have someone to help; 4) clearance to safely participate in the physical activity components of the intervention (Physical Activity Readiness Questionnaire (PARQ+ [19]) or written physician clearance for participation) and; 5) written informed consent to participate. Specific exclusion criteria for Phase 2 are: 1) best-corrected visual acuity of worse than 6/60 (legal blindness) in both eyes [20]; 2) unable to ambulate/walk for exercise; 3) serious illness (including mental and cognitive illness) limiting their ability to exercise or complete the trial; 4) contraindications to exercise (i.e., failure to fulfil the prerequisites of the PARQ+); 5) uncontrolled hypertension ($\geq$160/>90 mmHg); 6) signs or symptoms of alcohol/substance abuse and; 7) unable to commit to attending $\geq$80% of the sessions. Participation in Phase 1 of this study is not an exclusion criterion.

## Semi-structured interviews in Phase 1

The semi-structured interviews are estimated to take 15–20 minutes to complete (Table 1). The interview will include initial questions regarding: the participants' AMD diagnosis and its impact on their day-to-day life; and any previous experience participating in lifestyle programs and associated details. This will be followed by discussions about the proposed MINGLE program including comments on the duration, frequency and preferred time to schedule the sessions; the suitability of the planned exercises and AMD-related nutrition education topics; as well as any suggestions of content to include and/or exclude. The final interview questions will focus on motivation to participate in the program such as potential barriers reducing motivation and strategies to maintain motivation and commitment to completing the program.

## Intervention in Phase 2

Using Phase 1 findings, the proposed MINGLE program will be modified to better suit people with AMD. The proposed MINGLE program, adapted from WTL and the dietary telehealth intervention [17], will run for 1-hour per week for 10 weeks online via Zoom. Participants will be divided into subgroups based on participant preference to available session timings (e.g., morning/afternoon). An overview of the proposed MINGLE program has been summarised in Table 2. Briefly, each session will include 10 minutes of informal socialising; 30 minutes of physical activity; and 15 minutes of nutrition education. The physical activity session will not require any equipment other than a chair and will be facilitated by a research team member with an exercise-related qualification. Each physical activity session will include: 2-minute mindful breathing, 12-minute warm-up (e.g., stretching and bodyweight strengthening exercises), 7-minute moderate-to-vigorous intensity shadow boxing, and 9-minute cool down (e.g., bodyweight strengthening, balance exercises). The intensity of the exercises will be self-

**Table 1. Topic guide.**

| Topic | Phase 1 Interview | Phase 2 Focus group |
|---|---|---|
| Questions regarding previous program experience | Have you ever participated in a social program for seniors e.g., community-based program, exercise group, member of a club? If yes, what did the program involve and how did you find the program? | How did you the MINGLE program compare to other programs you have participated in? |
| Questions regarding the intervention design | What are your thoughts about the duration and frequency of the sessions? What would your preference be in terms of day and time if you were to participate in a program like this?. | What are your thoughts about the duration and frequency of the sessions? |
| | For exercise and nutrition components, other studies have included various exercises and nutrition topics. What are your thoughts on these? Do you have suggestions of what to include/ exclude or topics of discussion in the socialising component? If so, why? | How did you find the exercise and nutrition topics? Were there exercises or topics we should exclude or change? |
| | Do you think there is anything else that should be included in the MINGLE program? Do you think there is anything else that should be excluded from the MINGLE program? If so, why? | Do you think there is anything else we should have included/excluded in the MINGLE program? |
| Question regarding motivation (behaviour) | Could you describe any concerns you have about participating in the MINGLE program? | Did you have any concerns while participating in the MINGLE program? |
| | What do you think might be the best way to keep participants motivated and committed to the 10-week MINGLE program? | How could we have made the program more motivating and easier to commit to. |
| | Do you have any further comments regarding the MINGLE program or anything else we have spoken about today? | Do you have any other suggestions or feedback regarding the MINGLE program? |

evaluated using the Borg Rating of Perceived Exertion (RPE) Scale [21]. Modified versions of the exercises will also be demonstrated by the researcher for participants who find the exercises too easy or difficult. The AMD-related nutrition education will focus on evidence-based content on appropriate use of nutritional supplements (e.g., Age-Related Eye Disease Study (AREDS) and key food groups to incorporate as part of the habitual diet according to the Australian Dietary Guidelines [22]. The format of this nutrition educational content will alternate between presentations by the researcher and facilitated group discussions.

## Measures

At baseline, data on participants' AMD diagnosis such as stage and type of AMD, unilateral or bilateral AMD and details about any treatment received will be collected from the clinic. A pre-post intervention questionnaire will also be administered online via Research Electronic Data Capture (REDCap) to all participants involved in the intervention (Phase 2). The post-intervention questionnaire will be administered immediately after the 10-week intervention period. It will cover topics including: demographics (age, sex, ethnicity, education level, living situation (e.g., living alone or with others; homeowner or renter), marital status, employment status, and usual income); vision function; physical activity level; diet; falls self-efficacy; loneliness; quality of life; and program acceptability and feasibility. Specific instruments included in this questionnaire are described in more detail below.

**Table 2. Overview of the proposed MINGLE program.**

| Contents | | Duration (min) |
|---|---|---|
| Informal socialisation | Session 1: introduction of participants and facilitators; discussion of program expectations | 10 |
| | Sessions 2–10 catch up with participants regarding conversations from the previous session e.g., activities planned for the past week; recap of previous session including reflection on the exercises; nutrition topic. | |
| Physical activity component | Each session will include: 5 minutes of mindful breathing; a 15-minute warm-up and moderate intensity exercises (e.g., stretching and bodyweight strengthening exercises), and a 10-minute cool down (e.g., balance exercises and stretching). | 30 |
| Break | Post-exercise break to drink water and use facilities. | 5 |
| AMD-related nutrition education component | Five evidence-based nutrition topics (i.e., fruit and vegetables; fish and seafood; low glycaemic index foods; other important foods; and nutritional supplements for AMD) will be covered throughout the 10-week program. Each topic will include an educational presentation by the facilitator in one session, followed by a guided discussion on this topic in the following session such as sharing tips to incorporate the dietary recommendation. | 15 |
| | For example, Session 1 will include a presentation on fruits and vegetables including AMD-related benefits; recommended serving sizes according to the Australian Dietary Guidelines. Participants will then be encouraged to reflect on this information over the week. Session 2 will then focus on discussing potential barriers to consuming the recommended daily intake of fruits and vegetables and sharing tips to overcome these barriers. | |

1. The Active Australia Survey is a 13-item, self-administered questionnaire to measure the frequency and duration of recreational physical activity in the last week as well as knowledge about the health benefits of physical activity [23]. The knowledge component of the survey is presented as a five-point Likert scale with response options ranging from 'Strongly Disagree' to 'Strongly Agree' [23]. This instrument can be implemented via computer-assisted telephone interviewing or in face-to-face interviews and is the recommended Australian population-based physical activity survey [23, 24].

2. The Short Dietary Questionnaire for Age-related Macular Degeneration (SDQ-AMD) was used in the aforementioned telehealth intervention for people with AMD to capture dietary intake in the last week [25]. Response options to each food item is expressed as either serves consumed per day or per week, or number of times consumed per week [25]. The SDQ-AMD was adapted from a validated questionnaire [26] to capture intake of key food groups linked to AMD management. A question about dairy intake has also been added given evidence of a protective association between dairy intake and AMD [27]. The scoring criteria for the SDQ-AMD has been described previously [17] however will also be modified to provide a score for dairy intake.

3. National Eye Institute Visual Function Questionnaire-25 (VFQ-25) is a reliable and valid instrument to capture vision function in people with AMD [28]. This instrument contains questions on the following sub-scales: general health; general vision; expectations; wellbeing/distress; ocular pain; near vision; distance vision; peripheral vision; social function; colour vision; driving; role limitations; and dependency [29]. The VFQ-25 scores are reported as an average of the items within each sub-scale [29].

4. Falls Self-Efficacy Scale–International (FES-I) is a 16-item questionnaire to assess participants' concerns about their possibility of falling [30]. Response options are presented on a four-point Likert Scale ranging from 'Not at all concerned' to 'Very concerned'. Total scores range from 16 to 64 where a score of 16–19 indicates 'low concern'; 20–27 is 'moderate concern'; and 28–62 is 'high concern' [30].

5. De Jong Gierveld Loneliness Scale is valid 6-item questionnaire to reliably capture overall, emotional and social loneliness [31]. This scale was similarly used in our WTL-Z RCT among older adults living in the United Kingdom. Scores range from 0 (not lonely) to 6 (extremely lonely). The guide suggests that the following items should be reversed before scoring: "There are plenty of people I feel close to"; "There are many people I can completely trust"; and "There are enough people I feel close to" [31].

6. The Assessment of Quality of Life-8 Dimensions (AQoL-8D) is a 35-item questionnaire that includes eight individually scored dimensions [32]. The AQoL-8D comprises the 6 dimensions within the AQoL-6D (i.e., independent living; relationships; mental health (including depression); coping; pain; and senses) with two additional dimensions (i.e., self-worth; and happiness) [32]. The scoring guide for each dimension have been developed by the researchers at Monash University, Australia who developed the instrument (Assessment of Quality of Life, n.d.). In brief, the score ranges for each dimension are as follows: 4 to 22 (independent living); 7 to 34 (relationships); 8 to 41 (mental health); 3 to 15 (coping); 3 to 13 (pain); 3 to 16 (senses); 3 to 15 (self-worth); and 4 to 20 (happiness) [32].

7. The 'Attitude to Falls-Related Intervention Scale' (AFRIS) [33] will be administered pre-post intervention to assess participant acceptability of the program. This will be achieved by comparing the pre-post mean total scores of participants offered the intervention, with possible scores ranging from 6 to 42 i.e., 'less positive' to 'more positive' attitude towards the intervention [33].

In addition to the AFRIS, the following approaches will be used to assess the acceptability and feasibility of the MINGLE program. 1) A 30-minute feedback discussion to reflect on the MINGLE program will be held immediately after the tenth session of the program. Participants will be invited to join the discussion by remaining online after completing their tenth session. Participants will be asked to reflect on a range of topics including their program experience; suggestions on how to improve the program; and the program's potential for a large-scale roll-out. 2) A short feedback form will be included in the post-intervention questionnaire. This will include open- and close-ended questions covering level of satisfaction with the program, self-determined adherence to the program, and general feedback (e.g., suggestions to improve the program). 3) Attendance data will also be collected on all participants to allow for reporting on: the number sessions attended, completed sessions ('dose' of intervention received), and reasons for missed sessions. The session content will be tracked via checklists completed after each session allowing for reporting on the extent to which the intervention content is delivered per protocol and percentage of participants engaging/completing the activities.

## Outcome measurement

Data collected on the abovementioned measures will be used to evaluate the program's preliminary efficacy and its acceptability and feasibility. Specifically, this includes an evaluation of: 1) recruitment strategies–identifying the reach of each recruitment source; 2) recruitment rates at each stage of the project i.e., total number who expressed an interest in Phase 1 or 2,

proportion who were eligible, provided consent, and completed the baseline questionnaire; 3) program adherence based on self-reported adherence and the Research Assistant's evaluation during each session; 4) attrition rate; 5) pre-post responses to the AFRIS questionnaire; and 6) qualitative data obtained from the feedback survey and focus group discussion immediately post-intervention (Fig 1). The feedback survey includes Likert scales to rate overall satisfaction with the program and adherence to the nutritional advice provided, two Yes/No questions to determine whether participants would recommend the program to a friend and if the program was worthwhile, and two open-ended questions to identify the most useful or valuable components of the program, and for additional suggestions to improve the program.

Secondary outcome measures include participant-specific outcomes including a significant reduction in loneliness according to the De Jong Gierveld Scale, after completion of the 10-week MINGLE program, and significant changes in depression; quality of life; falls self-efficacy; physical activity (level) and overall diet using the AQoL-8D's mental health dimension, overall AQoL-8D, FES-I, Active Australia Survey, and SDQ-AMD, respectively.

## Sample size considerations

**Phase 1.** Participants for Phase 1 will be recruited until data saturation is achieved, that is, when the interviews no longer provide new information or knowledge to the researchers about an AMD patient's perspective on participating in the MINGLE program. According to our previous research investigating the barriers and facilitators to participating in a mental wellbeing program among people with AMD [34], approximately 30 participants were required to achieve this.

**Phase 2.** A minimum of 52 participants are required to achieve a pre-post change in the primary outcome (i.e., loneliness), assuming a 0.4 effect size according to Cohen's d and 80% power (G*Power 3.1 software). An effect size of 0.4 was chosen as the average effect size of our previous WTL-related work i.e., 0.43 [35] and 0.35 [16] for loneliness using the De Jong Gierveld Scale. The later study also proved to be feasible and acceptable demonstrating $\geq$ 80% adherence rate, 88% retention rate and was acceptable or highly acceptable by $\geq$ 95% of participants [16].

## Data collection and analysis plan

**Phase 1.** Semi-structured interviews will be conducted Data for all participants will be audio recorded and transcribed verbatim using an external transcribing service. Transcripts will be analysed iteratively to ensure analytic reflexivity. Two members of the research team with a vision science and/or ophthalmology background will deductively code the transcripts using content analysis [36] based on the Capability, Opportunity, Motivation and Behaviour (COM-B) model of the behaviour change wheel [37]. Codes will be discussed, and consensus reached with other members of the research team. Themes will be established under each subset of the COM-B model relating to barriers and facilitators to participating in an interactive online program. Phase 1 findings will inform the design of MINGLE.

**Phase 2.** Data for all participants will be collected via online self-report pre-post questionnaires administered via REDCap. Participants will be prompted to complete and return questionnaires by study staff to promote participant retention. All data will be entered into REDCap. Only staff directly involved in the study will have access to the databases. Statistical analysis will be carried out using IBM SPSS Statistics V.25. Descriptive statistics will be used to assess the acceptability and feasibility of MINGLE and the pre-post intervention changes in loneliness, depression, falls self-efficacy, physical activity, and dietary intakes. For missing data, the last observations will be carried forward prior to data analysis.

## Safety measures and end point

The physical activity researcher (RM) has extensive experience in conducting physical activity program evaluations in adults and children and the accredited practising dietitian (DT) has experience in conducting a telehealth nutrition intervention in older adults with AMD which included the provision of appropriate nutrition advice and counselling. The co-authors (NB and CJ) also have firsthand experience administering WTL and WTL-Z and will provide oversight on the study as well. The research team member facilitating the MINGLE program will have an exercise-related qualification and accreditation to practice. Rated a low-risk study, adverse events are unlikely to occur. However, safety endpoints will be indicated where participants report or present with persistent pain or injury after performing the exercises. In this case, appropriate documentation will be completed, and the participant will be excused from participating in the exercise sessions until they completely recover and advised to seek medical and/or allied health advice where appropriate.

## Patient and public involvement

This study will have patient involvement in the adaptation of the MINGLE program. Patient involvement will occur in Phase 1 of the study where semi-structured interviews are conducted to identify the facilitators and barriers to participating in an online lifestyle intervention program to help inform the development of MINGLE. Patients will not be involved in the recruitment or conduct of the program itself. Overall results will be disseminated to consenting participants at the end of the study as a one-page lay summary.

## Supporting information

**S1 File.**
(DOC)

**S2 File.**
(DOCX)

## Author Contributions

**Conceptualization:** Diana Tang, Rona Macniven, Charlotte Jones, Bamini Gopinath.

**Funding acquisition:** Diana Tang, Rona Macniven, Charlotte Jones, Bamini Gopinath.

**Methodology:** Diana Tang, Rona Macniven, Charlotte Jones, Bamini Gopinath.

**Writing – original draft:** Diana Tang, Rona Macniven, Nicholas Bender, Charlotte Jones, Bamini Gopinath.

**Writing – review & editing:** Diana Tang, Rona Macniven, Nicholas Bender, Charlotte Jones, Bamini Gopinath.

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
