## [Decision Letter · Decision Letter 0]

2 Feb 2022

PONE-D-21-26828

Development, implementation and evaluation of the online Movement, Interaction and Nutrition for Greater Lifestyles in the Elderly (MINGLE) program: the protocol for a pilot trial.

PLOS ONE

Dear Dr. Tang,

Thank you for submitting your manuscript to PLOS ONE. After careful consideration, we feel that it has merit but does not fully meet PLOS ONE’s publication criteria as it currently stands. Therefore, we invite you to submit a revised version of the manuscript that addresses the points raised during the review process.

We look forward to receiving your revised manuscript.

Kind regards,

Walid Kamal Abdelbasset, Ph.D.

Academic Editor

PLOS ONE

https://journals.plos.org/plosone/s/file?id=ba62/PLOSOne_formatting_sample_title_authors_affiliations.pdf”

“This work is supported by the Macular Disease Foundation Australia Research Grant Award.”

“D.T, R.M, C.J, and B.G were awarded the Macular Disease Foundation Australia Research Grant by the Macular Disease Foundation Australia to conduct this research project.

https://www.mdfoundation.com.au/

The funders had and will have a role in the study design, data collection and analysis, decision to publish or preparation of the manuscript.”

Reviewers' comments:

Reviewer's Responses to Questions

**Comments to the Author**

1. Does the manuscript provide a valid rationale for the proposed study, with clearly identified and justified research questions?

Reviewer #1: Yes

Reviewer #2: Yes

2. Is the protocol technically sound and planned in a manner that will lead to a meaningful outcome and allow testing the stated hypotheses?

Reviewer #1: Partly

Reviewer #2: Yes

3. Is the methodology feasible and described in sufficient detail to allow the work to be replicable?

Reviewer #1: No

Reviewer #2: Yes

4. Have the authors described where all data underlying the findings will be made available when the study is complete?

Reviewer #1: Yes

Reviewer #2: Yes

5. Is the manuscript presented in an intelligible fashion and written in standard English?

Reviewer #1: Yes

Reviewer #2: Yes

6. Review Comments to the Author

You may also provide optional suggestions and comments to authors that they might find helpful in planning their study.

Reviewer #1: The project is well thought of and can provide significant support for people with AMD. However, I have major and minor concerns about the protocol.

First, the Qualitative component of the project as a sequential exploratory design might not provide clear guidance to the quantitative part. The project is looking for specific answers regarding MINGLE program ( duration, frequency..etc.); however, participants do not know anything about it. Therefore, it is unlikely that their input will help in structuring phase2.

2- Additionally, the analysis section of Phase1 needs further information about who will conduct the interviews(background and qualification), where the interview will be held, who will do the thematic analysis, and whether it is inductive or deductive. Moreover, it is advisable to provide a topic guide for the semi-structured interview.

3- There is no information on the minimal visual requirement in the inclusion criteria since you use a video conferencing platform (Zoom).

4- the aim of the project was (feasibility, acceptability, and efficacy). While the "Attitude to Falls-Related Intervention Scale" could provide quantitative information about the "acceptability," I am not sure it could guide feasibility. Can you state specific parameters that can determine the feasibility for a larger study; like attrition rate, follow-up and qualitative data regarding length of the study, outcome measure appropriateness (e.g., length).

minors:

1- use APA style to cite "ZOOM"

2- Bender et al. should not be cited in the main text. use APA style to cite it in the reference section.

Reviewer #2: - The physical activity component needs more details as what type of exercises will be done and how would you determine the difficulty of the exercises!

7. PLOS authors have the option to publish the peer review history of their article (what does this mean?). If published, this will include your full peer review and any attached files.

Reviewer #1: No

Reviewer #2: **Yes: **Saud F. Alsubaie

---

## [Author Response · Author response to Decision Letter 0]

25 Feb 2022

Please see attached Resubmission Cover Letter and Response to Reviewer documents for responses to comments related to this submission.

---

## [Decision Letter · Decision Letter 1]

12 Apr 2022

Development, implementation and evaluation of the online Movement, Interaction and Nutrition for Greater Lifestyles in the Elderly (MINGLE) program: the protocol for a pilot trial.

PONE-D-21-26828R1

Dear Dr. Tang,

We’re pleased to inform you that your manuscript has been judged scientifically suitable for publication and will be formally accepted for publication once it meets all outstanding technical requirements.

Kind regards,

Walid Kamal Abdelbasset, Ph.D.

Academic Editor

PLOS ONE

Additional Editor Comments (optional):

Reviewers' comments:

Reviewer's Responses to Questions

**Comments to the Author**

1. Does the manuscript provide a valid rationale for the proposed study, with clearly identified and justified research questions?

Reviewer #1: Yes

Reviewer #2: Yes

2. Is the protocol technically sound and planned in a manner that will lead to a meaningful outcome and allow testing the stated hypotheses?

Reviewer #1: Yes

Reviewer #2: Yes

3. Is the methodology feasible and described in sufficient detail to allow the work to be replicable?

Reviewer #1: Yes

Reviewer #2: Yes

4. Have the authors described where all data underlying the findings will be made available when the study is complete?

Reviewer #1: Yes

Reviewer #2: Yes

5. Is the manuscript presented in an intelligible fashion and written in standard English?

Reviewer #1: Yes

Reviewer #2: Yes

6. Review Comments to the Author

You may also provide optional suggestions and comments to authors that they might find helpful in planning their study.

Reviewer #1: I Think the authors made an outstanding amendment to the protocol. All comments were addressed properly. Good luck with your study.

Reviewer #2: Dear author

I appreciate your effort in revising the manuscript, and thank you for considering all the comments sent to you.

7. PLOS authors have the option to publish the peer review history of their article (what does this mean?). If published, this will include your full peer review and any attached files.

Reviewer #1: **Yes: **Ahmed Alhowimel

Reviewer #2: No

---

## [Editor Report · Acceptance letter]

27 Apr 2022

PONE-D-21-26828R1 

Development, implementation and evaluation of the online Movement, Interaction and Nutrition for Greater Lifestyles in the Elderly (MINGLE) program: the protocol for a pilot trial. 

Dear Dr. Tang:

I'm pleased to inform you that your manuscript has been deemed suitable for publication in PLOS ONE. Congratulations! Your manuscript is now with our production department. 

Kind regards, 

on behalf of

Dr. Walid Kamal Abdelbasset 

Academic Editor

PLOS ONE